# Study on Protective Coal Pillar Size Design for Ultra High Voltage Line Tower Mining in Mountain Areas

**Feiya Xu [1],\*** , **Wenbing Guo [1,2]** and **Jianli Li [3]**

1   School of Energy Science and Engineering, Henan Polytechnic University, Jiaozuo 454000, China; guowb@hpu.edu.cn

2   Collaborative Innovative Center of Coal Safety Production in Henan Province, Henan Polytechnic University, Jiaozuo 454000, China

3   Sihe Mine, Jincheng Anthracite Mining Group, Jincheng 048205, China; jily1028@126.com

\*   Correspondence: 111502010002@home.hpu.edu.cn; Tel.: +86-159-7878-5114

**Abstract:** High voltage line towers in mining areas are sensitive to surface deformation caused by mining. Protective coal pillar design for high voltage towers is one of the commonly-used technical measures. Aiming to solve the coal mining safety problem under the Ultra High Voltage transmission line in Sihe Coal Mine of Shanxi Province, the angle and size of protective coal pillars with the vertical line method were analyzed in this paper. The effect of additional displacement caused by landslide or slippage mining in mountain areas and repeated mining was considered. Based on the principle of the vertical line method, the protective coal pillar range and size were calculated. The amount of coal deposited in coal pillars for high voltage line towers was compared and analyzed between the vertical line method and the linear structure method. The results showed that the angle of critical deformation decreased by 2~10° caused by slippage due to mining in a mountainous area, and the angle in the uphill direction of building decreased more than that in the downhill direction; when multi-seams were mined repeatedly, the angle of critical deformation in the lower seam coal mining was reduced by 5~10° compared with that of the upper seam. The protective coal pillar design with the vertical line method can protect the high voltage line towers more effectively, and the amount of protective coal pillars with the vertical line method was 5.8 million tons less, which avoided the waste of coal resources.

**Keywords:** the Ultra High Voltage transmission line tower; slippage; mining in mountain area; the vertical line method; protective coal pillar; the angle of critical deformation

## 1. Introduction

As the main infrastructure for digesting coal resources and long-distance power transmission in mining areas, the high voltage transmission line towers will inevitably pass through coal mining areas, resulting in a large amount of coal resources being pressed and unable to be exploited [1–3]. In view of the special structure and importance of high-voltage towers, how to mine coal safely under a high-voltage tower and protect it is a problem that scholars around the world have been working on [4–12]. Early in the 1970s, the material of aggregate backfilled into the gob method was implemented by American scholar T. W. Bernett [13] to enhance the stability of tower foundation and ensure the safe operation of high-voltage tower in the Pittsburgh mining area, to solve the problem of mining under a Ultra High Voltage line of 345 kV. R. W. Bruhn et al. [14] studied the influence of surface movement and deformation on high-voltage towers with steel structures. Guo et al. [15–17] analyzed and calculated the displacement deformation and additional stress of the high-voltage transmission tower under mining influence combined theoretical model with field measurement.

It concluded that the resistance of the high-voltage transmission tower to surface deformation was greater than that of the general building (structure). Liu and Gao [18] used FLAC 3D numerical simulation (which was a numerical simulation software used in Geology, underground engineering, geological simulation et al.) method to analyze the stability of the foundation of the tower in the mining-affected area, and found that the foundation inclination would endanger the stability of the Ultra High Voltage (UHV) transmission line. In order to recover as much coal resources as possible, Yan and Dai [19] took the extra- thick and steep inclined coal seam mining in Huating mine of Gansu Province as the research object, and carried out the optimum design of the setup entry of the first panel for the high-voltage tower in the mining-affected area. In the study of the characteristics of surface deformation in mountainous areas, He [20] analyzed the superposition principle of surface slip movement in mountainous areas, and established a mathematical model for predicting surface movement in mountainous areas. On the basis of the data of surface movement observation stations in major mining areas of Shanxi Province, Han [21] et al. studied the prediction parameters of the surface slip model in mountain areas, and expanded the application scope of the mining subsidence prediction model in mountain areas. Qin et al. [22] believed that mining in mountain areas was closely related to the location of surface buildings. When buildings were located in the valley or inflection point, the additional deformation caused by mining was more serious and the buildings were very vulnerable to damage. Based on four panels of data in the Jincheng mining area, Feng [23] revealed that the sliding by mining in mountainous area increased the range of movement in the uphill direction, thus reducing the angle of critical deformation, and the amount of reduction of the angle was different in different slopes.

Through the analysis of the literature at home and abroad, it was shown that the research mainly focused on theoretical analysis and technical maintenance for high-voltage tower movement and deformation. Therefore, during the actual mining of a coal mine, the protective coal pillar size for the high-voltage tower was the issue of most concern. At present, there has been no detailed research and analysis on this aspect. Based on a case study under the Ultra High Voltage line towers in mining-affected areas, this paper analyzed the angle of critical deformation of strata parameters under different mining effects in this mine and the protective coal pillars for Ultra High Voltage line towers with different methods. It was concluded that the protective coal pillars design for Ultra High Voltage line towers with the vertical line method was more accurate and reasonable to effectively protect the safe operation of the Ultra High Voltage line and the towers on the ground while maximizing coal resources extraction.

## 2. Geological Mining Conditions and the Ultra High Voltage Towers

Sihe Coal Mine is located in the North-west of Jincheng City, Shanxi Province, China. It is divided into two mining areas, the West area and the East area, as shown in Figure 1. The study area of the paper is located in the East area of Sihe Coal Mine. The coalfield area is 114.5 km$^2$ with 15 coal seams, of which No. 3 and No. 15 are the main workable seams, No. 9 is a partially workable seam (most of the eastern area was workable), and the rest are the non-workable seams, as shown in Table 1. Fully-mechanized full seam coal mining technology is employed. The seams are nearly horizontal and generally not affected by geological tectonic activities.

The mining-affected areas of Shanxi section—Sihe Coal Mine were selected as the research object in this paper, in which the protective coal pillar design for the UHV Transmission Line Tower was studied. The total length of this section was about 120 km. It passed through a large mining area with a large number of coal sources and complex underground mining conditions of Sihe Coal Mine, as shown in Figure 2. Among them, seam #3 was being mined on the east of line tower No. N143~N147 where seam #15 was on the east of N152~N157 and the north of N157~N167 (about 83.9 m distance above the seam #3 which had been mined). The UHV Transmission Line Towers were self-supporting iron towers presented as the "wine cup" type, which was about 83.4m high (the distance from the top to the foundation of the towers), as shown in Figure 3a. Each tower was connected by steel wires to

form a "highway" linear structure. The tower was supported by four separated foundations with a distance of 25 m (root), all of which were grouted with concrete, as shown in Figure 3b,c.

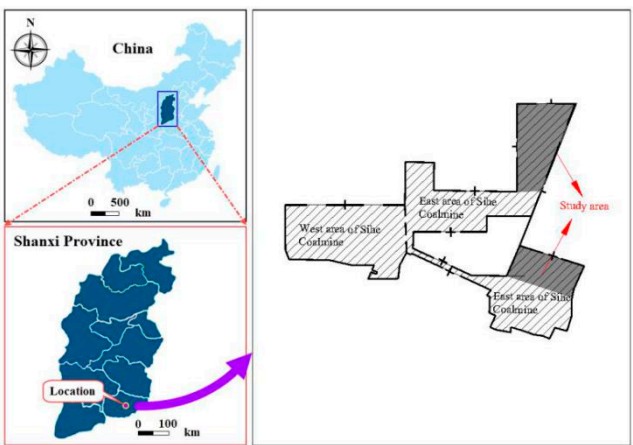

**Figure 1.** Geographic map of study area location.

**Table 1.** Characteristic of workable coal seam in Sihe coalfield.

| Stratum | Coalsea (m) | Coal Seam Thickness Minimum–Maximum Average (m) | Seam Spacing Minimum–Maximum Average (m) | Workability | Roof and Floor | |
|---|---|---|---|---|---|---|
| | | | | | Roof | Floor |
| Shanxi Formation | 3 | 5.74–7.45 6.31 | 43.00–59.28 47.57 | Workable | Siltstone Sandy mudstone | Black mudstone Sandy mudstone Siltstone |
| Taiyuan Formation | 9 | 0–2.37 1.34 | 24.15–2.15 36.38 | Most were workable. | Siltstone | Mudstone Siltstone |
| | 15 | 2.02–5.45 2.69 | | Workable | Limestone | Mudstone |

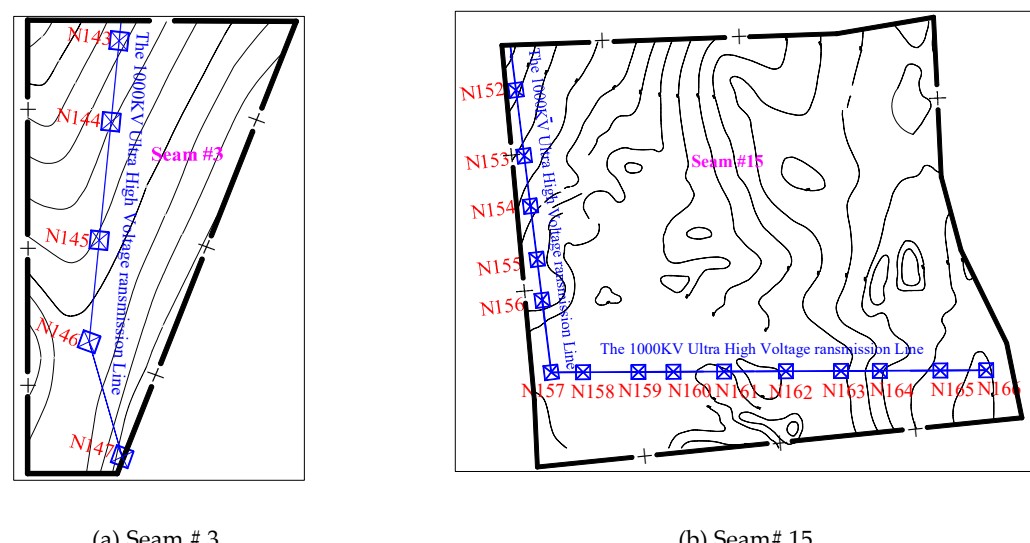

(a) Seam # 3          (b) Seam# 15

**Figure 2.** Location of the Ultra High Voltage line tower layout in Sihe coalmine.

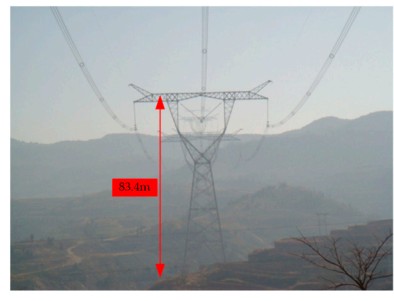

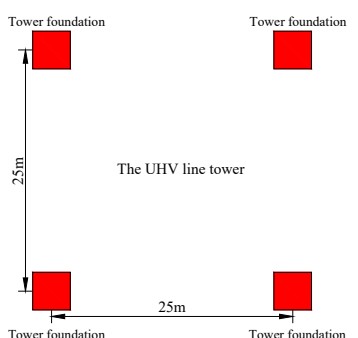

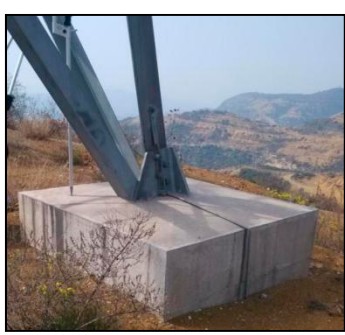

(a) The high of the towers

(b) Schematic diagram of tower foundation

(c) One of the Foundations of the UHV Tower

**Figure 3.** Construction and characteristic of the Ultra High Voltage line tower.

## 3. Selection and Analysis of the Angle of Critical Deformation of Strata

The UHV Transmission Line protection level was level I and its safety berm width was 20 m [1], as shown in Table 2. The angle parameter of protective coal pillar was selected based on the angle of critical deformation of strata [24].

**Table 2.** Classification of protection level and safety berm width of structures in mining area.

| Protection Level | Structures | Safety Berm Width/m |
|---|---|---|
| I | Expressways, UHV transmission towers, large tunnels, trunk line of oil (gas) pipeline, main ventilator rooms in mines, etc. | 20 |

### 3.1. Selection of the Angle of Critical Deformation by P Coefficient Method

The P coefficient method was based on the lithology of overlying strata to select the predicted parameters of the probability integral method and strata movement parameters in the mining area [1,16], as shown in Table 3.

**Table 3.** The angle of critical deformation selection according to the lithology of overlying strata.

| The Lithology of Overlying Strata | The Angle of Critical Deformation/(°) | | |
|---|---|---|---|
| | Strike Direction ($\delta$) | Head End ($\gamma$) | Tail End ($\beta$) |
| Hard | 75~80 | 75~80 | $\delta - (0.7\text{\textasciitilde}0.8) \times \alpha$ |
| Medium hard | 70~75 | 70~75 | $\delta - (0.6\text{\textasciitilde}0.7) \times \alpha$ |
| Weak | 60~70 | 60~70 | $\delta - (0.3\text{\textasciitilde}0.5) \times \alpha$ |

Note: The strike direction means the angle along the seam strike direction; head end means the angle towards the head end direction; tail end means the angle towards the tail end direction [25].

Based on the geological and mining conditions of Sihe Coal Mine and the borehole column diagrams of coal seams #3 and #15, the comprehensive evaluation coefficient P of overlying strata in this area was 0.715, and the lithology of overlying strata was weakly inclined to medium-hard. According to Table 3, the angle of critical deformation was selected as follows: the strike direction angle was 65°, the head end angle was 63°, and the tail end angle was 65°.

### 3.2. Selection of the Angle of Critical Deformation by Repeated Mining

After repeated mining, the value of surface deformation would be larger than that of the first mining, thus the discontinuous damage of the surface would also increase. Therefore, large cracks or step damage would occur, and the angle parameters of surface movement would change. According

to the analysis of large observation data [1], the angle of critical deformation would decrease by 5~10° during the mining process. Because the High Voltage line tower belonged to the towering structure, its deformation resistance was greater than that of the common masonry structure, so the angle of critical deformation would decrease by 5° [15,16]. Therefore, under the repeated mining conditions in coal seam #15 of Sihe Coal Mine, the angle of critical deformation was selected as follows: the strike direction angle was 60°, the head end angle was 58°, and the tail end angle was 60°.

*3.3. Analysis of the Angle of Critical Deformation by Considering Additional Influence of Slippage Mining in Mountain Area*

In addition to the movement caused by mining, the additional movement caused by landslide or slippage will also occur, which will lead to an increase in the range of surface movement in mountain areas, and consequently damage the buildings (structures) located within the influence range of slippage [26,27].

Generally, the slippage profile was assumed to be a curve. The slippage vector at any point can be decomposed into tangential component p and normal component q, which can be decomposed into vertical component w (subsidence) and horizontal component u (horizontal displacement) in response to surface movement, as shown in Figure 4. In point A of a convex landform, the superposition of the vertical component caused by the slope slippage and the subsidence caused by mining will occur, which increases the surface subsidence; in point B of a concave landform, the offset of that will occur, which decreases the surface subsidence. The horizontal component caused by slippage, that is, horizontal displacement, always pointed the tail end, which enlarged the range of surface movement.

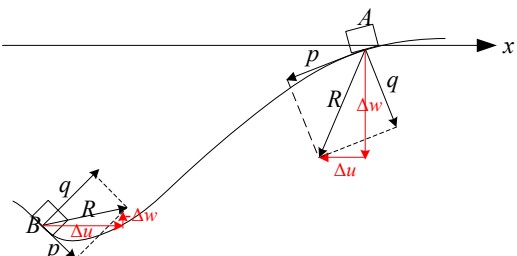

**Figure 4.** Additional direction analysis of slope slippage.

Taking the coal mining under the UHV Line Towers in Sihe Coal Mine as an example, the effect of slippage on the angle of critical deformation of strata was analyzed in Figure 5. The surface horizontal displacement u caused by underground mining always pointed to the center of the gob area. The horizontal component ($-\Delta u$) of the tower No. 1 in the downhill direction caused by slippage was opposite to the surface horizontal displacement direction, which formed an "offset"; while the horizontal component ($\Delta u$) of the tower No. 2 and 3 in the uphill direction caused by slippage was the same as the surface horizontal displacement direction, which formed a "superposition". Therefore, the range of surface movement caused by slippage will increase, resulting in the decrease of the angle of critical deformation. The angle value in the uphill direction of building (structure) decreased more than that in the downhill direction, as shown in Formula (1).

$$\begin{aligned} \delta' &= \delta - \Delta_d \\ \delta'' &= \delta - \Delta_u \\ \Delta_d &< \Delta_u \end{aligned} \tag{1}$$

where: $\delta'$ is the downhill angle of critical deformation in mountain area, °; $\delta$ is the angle of critical deformation, °; $\delta''$ is the uphill angle of critical deformation in mountain area, °; $\Delta_u$ is the corrected value of the uphill angle of critical deformation in mountain area, °; and $\Delta_d$ is the corrected value of the downhill angle of critical deformation in mountain area, °.

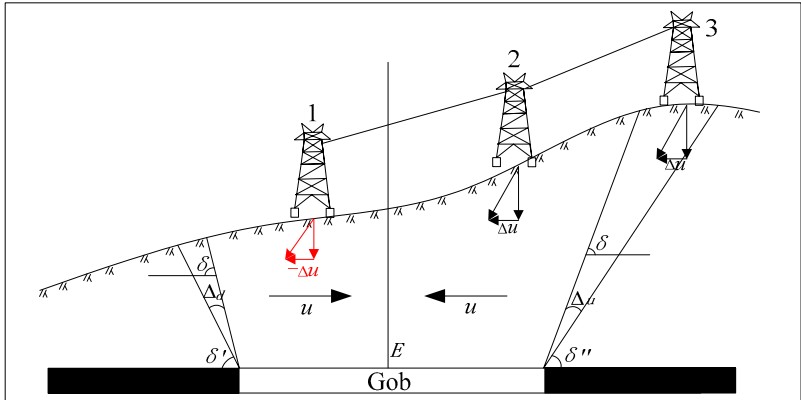

**Figure 5.** Effect analysis of slippage on the angle of critical deformation of strata.

According to "Standards for coal pillar design and coal mining under buildings, water bodies, railways and main roadways" [1]: The angle of critical deformation in a mountain area located in the uphill direction of the building (structure) should be reduced less by 5–10°, and the downhill direction should be reduced less by 2–3°. Therefore, the angle of critical deformation in a mountain area of the protective coal pillar design for the UHV Line Towers is shown in Table 4.

**Table 4.** Angle parameter of protective coal pillar design for the high voltage line tower (taking into account the additional influence of slippage caused by mining in mountain area).

| Coal Seam | | #3 | | | #15 (Repeated Mining) | |
|---|---|---|---|---|---|---|
| The angle of critical deformation | | Additional influence of slippage caused by mining in mountain area | | | Additional influence of slippage caused by mining in mountain area | |
| | | The uphill direction | The downhill direction | | The uphill direction | The downhill direction |
| The strike direction (°) | 65 | 60 | 63 | 60 | 55 | 58 |
| The tail end direction (°) | 63 | 58 | 61 | 58 | 53 | 56 |
| The head end direction (°) | 65 | 60 | 63 | 60 | 55 | 58 |

## 4. Analysis of the Protective Coal Pillar Size

### 4.1. The Method of Protective Coal Pillar Design

#### 4.1.1. The Vertical Line Method

The vertical line method was a method that combined graphics with calculation to design protective coal pillars. Taking the UHV line tower as an example, the steps were as follows:

Determination of the Boundary of Protected Area

In the mining and excavation plan, the outer boundary of four towers foundation of the UHV line tower was connected into a square as the boundary of the protected area, and then the safety berm was drawn around the boundary line. The safety berm width d was obtained, as shown in Figure 6.

Determination of Protective Boundary of Unconsolidated Formation

A distance of $s$ from the boundary of the protected area was the protective boundary of unconsolidated formation, which was calculated by Formula (2):

$$s = h \cdot \cot \varphi \tag{2}$$

where: $s$ is the protective boundary width of unconsolidated formation, m; $h$ is the unconsolidated formation thickness, m; and $\varphi$ is the angle of critical deformation of unconsolidated formation.

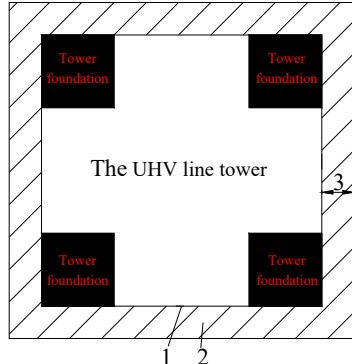

**Figure 6.** Determination of the boundary of protected area with the vertical line method. 1—The Ultra High Voltage (UHV) line tower boundary; 2—Safety berm; 3—Safety berm width.

Determination of Protective Coal Pillar Boundary

On the basis of step (2), the vertical line of the protective boundary was drawn from the corner points of the protective boundary of unconsolidated formation, and the length $q$ was taken when the vertical line pointed to the head end direction, and the length l was taken when the vertical line pointed to the tail end direction. The length of $q$ and $l$ were calculated by Formulas (3) and (4) respectively:

$$q = \frac{(H - h)\cdot \cot \beta'}{1 + \cot \beta'\cdot \tan \alpha \cdot \cos \theta} \tag{3}$$

$$l = \frac{(H - h)\cdot \cot \gamma'}{1 + \cot \gamma'\cdot \tan \alpha \cdot \cos \theta} \tag{4}$$

where: $H$ is the coal seam depth of the calculating point, m; $h$ is unconsolidated formation thickness, m; $\theta$ is the acute angle between the protected boundary and the coal seam in strike direction, °; $\alpha$ is coal seam dip, °; and $\beta'$, $\gamma'$ is the oblique crossing angle of critical deformation in oblique crossing section, °.

Determination of the Range of Protective Coal Pillar

According to the calculated $q$ and $l$ length, the vertical lengths were intercepted and connected to each point in turn and intersected in I, II, III and IV. Quadrilateral I II III IV was the protective coal pillar range set by the vertical line method, as shown in Figure 7.

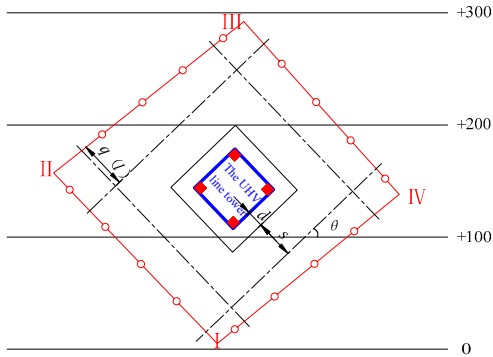

**Figure 7.** Diagram of protective coal pillar with the vertical line method.

Determination of the Protective Coal Pillar Size

According to the Figure 7 above, the calculation formula of the protective coal pillar length on one side of the UHV line tower was Formula (5):

$$W = d + s + q(l) \tag{5}$$

where: $d$ is safety berm width, m; $s$ is the protective boundary width of unconsolidated formation, m; and $q$ ($l$) is the vertical length of head end (tail end) direction, m.

### 4.1.2. The Linear Structure Method

The principle and calculation process of protective coal pillar design with the linear structure method were nearly the same as that of the vertical line method. The difference was that the boundary of the protected area was no longer delimited by a single high-voltage tower, but that all high-voltage towers were connected to form a straight line group, and then the boundary of the protected area was made. According to the average buried depth of the coal seam where the tower was located, the protective coal pillar was calculated. Based on the maximum protective coal pillar size, the parallel line of the protective boundary was made, as shown in Figure 8.

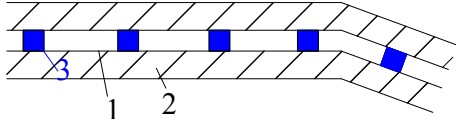

**Figure 8.** Determination of the boundary of protected area with the linear structure method. 1—The UHV line tower group boundary; 2—Safety berm; 3—The UHV line tower.

In addition, the limitations of the two methods presented should not be ignored, including that Formula (3) and (4) in step 3 of Section 4.1.1 were easily calculated but a lot of repeated calculation was needed. It should be careful to collect and analyze the data.

### 4.2. Result of Protective Coal Pillar Design

### 4.2.1. Result of Protective Coal Pillar Size

According to the analysis of the angle of critical deformation and calculation method of protective coal pillars, the vertical line method and linear structure method were used respectively to design protective coal pillars for a total of 20 towers in the east mine area of Sihe Coal Mine. The protective coal pillar size for each tower was obtained, as shown in Table 5.

**Table 5.** Results of protective coal pillar size for the UHV line tower.

| Coalseam | No. of the UHV Line Tower | Protective Coal Pillar Size(m) | |
|---|---|---|---|
| | | The Vertical Line Method | The Linear Structure Method |
| #3 | N143 | 361.6 | 334.6 |
| | N144 | 341.9 | 350.0 |
| | N145 | 233.5 | 228.7 |
| | N146 | 230.9 | 214.6 |
| | N147 | 166.4 | 187.3 |
| #15 | N152 | 347.2 | 358.2 |
| | N153 | 337.0 | 332.6 |
| | N154 | 348.3 | 314.3 |
| | N155 | 329.6 | 330.3 |
| | N156 | 223.2 | 236.5 |
| | N157 | 182.9 | 201.3 |
| | N158 | 163.7 | 171.6 |
| | N159 | 166.7 | 168.5 |
| | N160 | 175.2 | 185.6 |
| | N161 | 229.1 | 225.6 |
| | N162 | 232.3 | 214.9 |
| | N163 | 259.2 | 265.3 |
| | N164 | 494.6 | 359.3 |
| | N165 | 250.8 | 236.9 |
| | N166 | 208.4 | 201.6 |

According to Section 4.1, the shape of the protective coal pillar range obtained by the two methods was different. The range obtained by the vertical line method for each line tower was a quadrilateral, while the range obtained by the linear structure method was a line parallel to the protected boundary delineated by the maximum size of the protective coal pillar, as shown in Figure 9.

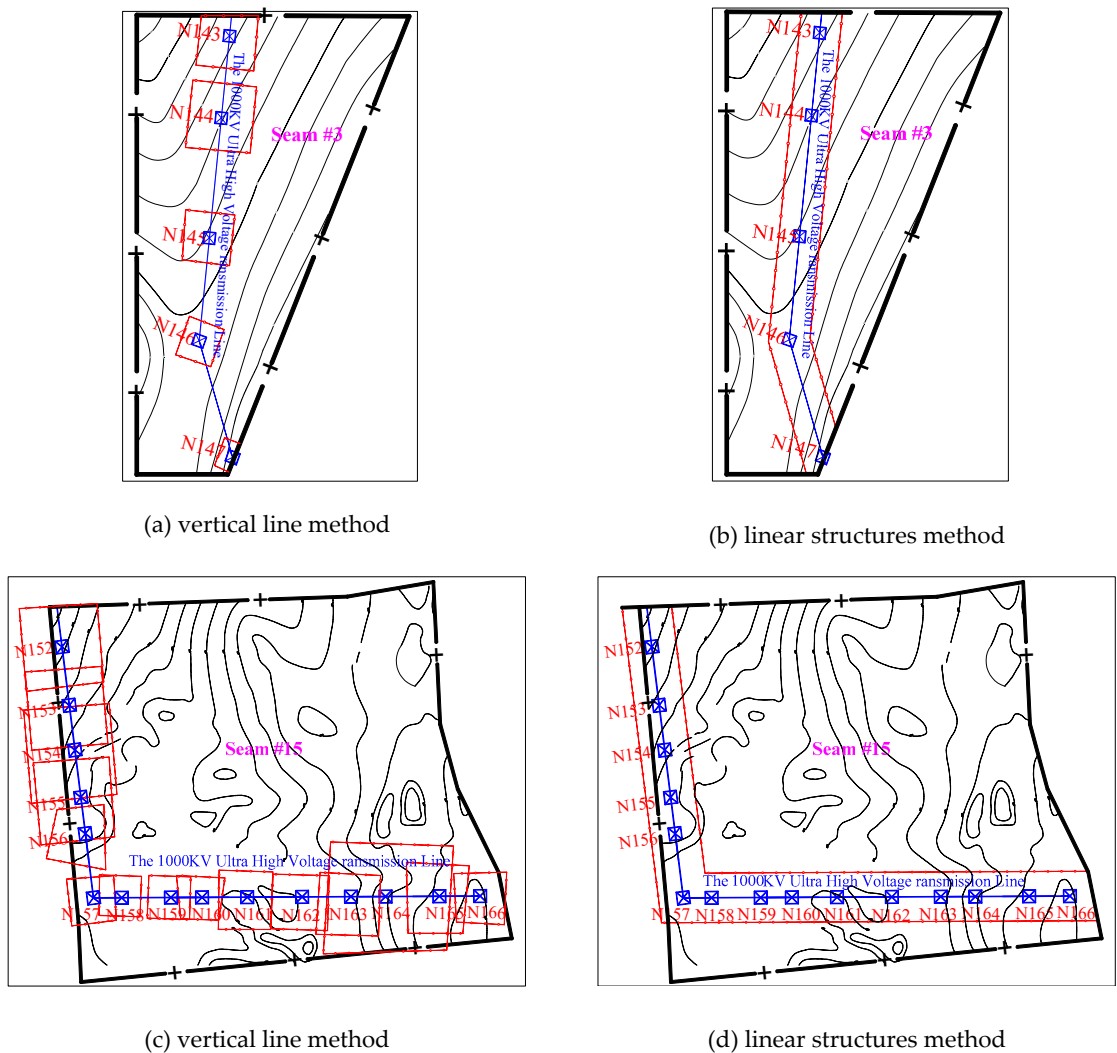

(a) vertical line method

(b) linear structures method

(c) vertical line method

(d) linear structures method

**Figure 9.** Schematic diagram of protective coal pillar design for an Ultra High Voltage line tower.

4.2.2. Result of the Protective Coal Pillar Amount

The protective coal pillar amount was calculated by Formula (6):

$$Q = V \times \gamma = \frac{A}{\cos \alpha} \times m \times \gamma \tag{6}$$

where: $Q$ is the protective coal pillar amount, t; $A$ is the projection of protective coal pillar area on the plane view, m2; $\gamma$ is the bulk density of coal, t/m3; $m$ is the thickness of the coal seam, m; $\alpha$ is the coal seam dip angle, °.

According to the range of protective coal pillar defined in Figure 8, the area and amount of protective coal pillars designed for the UHV line tower with the vertical line method and linear structure method were obtained respectively by Formula (6), as shown in Table 6.

**Table 6.** Comparison of the area and amount of protective coal pillar with different methods.

| Coal Seam | Thickness of Coal Seam (m) | Bulk Density of Coal (t/m$^3$) | Linear Structure Method | | The Vertical Line Method | |
|---|---|---|---|---|---|---|
| | | | The Area of Protective Coal Pillars (m$^2$) | The Amount of Protective Coal Pillars (t) | The Area of Protective Coal Pillars (m$^2$) | The Amount of Protective Coal Pillars (t) |
| #3 | 6.31 | 1.46 | 3,878,231.9 | 35,728,600 | 3,248,604.3 | 29,928,100 |
| #15 | 2.69 | 1.49 | 5,172,020.7 | 20,729,976 | 4,440,632.7 | 17,798,500 |

In Figure 9 and Table 6 it can be seen that the total area of protective coal pillar with the vertical line method was 1,361,015.6 m$^2$ less than that with linear structure method, and the total amount of protective coal pillars with the vertical line method was 8.7 million tons less, which created huge economic benefits. Because of the different seam depth of each UHV line tower and the additional influence of slippage caused by mining in a mountainous area, the protective pillars for each UHV line tower will be different. In addition, according to Figure 9, the range obtained by the vertical line method for each line tower was a quadrilateral. It meant that the protective coal pillar area depended on every individual quadrilateral area. However, the range obtained by the linear structure method was a line parallel to the protected boundary delineated by the maximum size of the protective coal pillar, which resulted in a waste of large amount of coal resources. Therefore, it was more reasonable and accurate to use the vertical line method to design protective coal pillars, which can achieve access to as much of the coal resources as possible under the premise of ensuring the safe operation of the UHV transmission line.

## 5. Conclusions

Based on a case study under Ultra High Voltage line towers in mining-affected areas, this paper analyzed the angle of critical deformation of strata parameters under different mining effects in this mine and the protective coal pillars for Ultra High Voltage line towers with different methods. It was concluded that:

The additional displacement caused by landslide or slippage in mountain areas resulted in an increase in the range of surface movement during the mining process and a decrease by 2–10° in the angle of critical deformation, and the angle value in the uphill direction of building (structure) decreased more than that in the downhill direction.

The protective coal pillar range design with the vertical line method was a quadrilateral corresponding to each UHV line tower respectively. It was more reasonable and accurate than the straight line range with the linear structure method, which not only avoided the waste coal resources waste, but also protected the UHV line tower more effectively.

The protective coal pillar design with the vertical line method can realize the coexistence of economic benefits and safety of the UHV line tower. This paper provided a reference for the panel layout in this mine area and had a certain guiding significance for the protective coal pillar design for a high voltage line tower in similar mine areas.

**Author Contributions:** W.G. conceived this paper and made some modifications; F.X. analyzed the data and calculate the results; J.L. provided the original materials; F.X. wrote the paper.

**Funding:** This research was funded by the National Natural Science Foundation of China, grant number 51774111.

**Acknowledgments:** The authors wish to thank the National Natural Science Foundation of China (project No. 51774111) for the financial support of this study and all the original data provided by Jincheng Anthracite Mining Group.

**Conflicts of Interest:** The author declares no conflict of interest.

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
