# Peer review of "Study on Protective Coal Pillar Size Design for Ultra High Voltage Line Tower Mining in Mountain Areas"

_designs_

Round 1
Reviewer 1 Report
The coal mining causes ground movements and deformations on the surface. These movements can influence the building structures and technical infrastructure and can damage them as well. The reviewed manuscript deals with the mining influences on the ultra high voltage towers in China. The case study presented in the manuscript can be interesting because the whole thickness of the coal seams is about 10 m and the mining has been done fully automatic. Author provide the modeling of the angle of critical influence based on several methods and rules. The methods used in the manuscript are not novel. But some novelty can be found in the consideration of additional effects, which influences on the angle of the critical deformation, namely the slippage and landslide caused by the slope in mountain area. Then Author calculated the border of the safety area on the surface, and the protective coal pillar. The coal pillar was designed with the use of two methods. The differences of the whole volume in the coal pillar has been shown for the both methods in the last chapters. That is the pity that the theoretical calculations was not compare with the measurements (monitoring results) in situ.
There are more known methods for designing of protective pillars for shafts and objects on the world. It would be interesting to make more literature research for improving the international relevance.
The design of the pillar in the vertical cross-section would be much clearer for the readers. That is why the additional figure would be improve the quality of the paper.
The novelty of the presented method is rather poor. On the other hand the presented results are significant and the presented equations can be important for the other test areas.
Following are my particular remarks:
1) what is the high are the towers ? There are only the description of the distance of separate foundation.
2) Fig. 3 – please change the description in English,
3) It is not clear why the vertical line method is more accurate than the other one ? Please describe the reasons more clearly,
4) Author have not mentioned any limitations of the presented methods. Please consider this remark.
5) The statement in the rows 260-262 are obvious. It is difficult to see the reasons given in the last sentence in this chapter (rows 262-264).
Author Response
Dear Reviewer:
I appreciate it very much to receive your comments on my paper. I have revised the manuscript according to the reviewers' comments one by one carefully, as shown:
Point 1: what is the high are the towers ? There are only the description of the distance of separate foundation.
Response 1: I called the third author Jianli Li who worked in the Sihe Coal Mine. He said the high of all the towers in the study area is about 83.4m. I have revised the manuscript in Figure 3(a).
Thanks!
Point 2: Fig. 3 – please change the description in English
Response 2: I am very sorry not to explain the Figure 3. According to the reviewers' comments, the description of Figure 3 is :
The UHV Transmission Line Tower were self-supporting iron towers presented as “wine cup” type, which was about 83.4m high (the distance from the top to the foundation of the towers ), as shown in Figure 3(a). Each tower was connected by steel wires to form a "highway" linear structure. The tower was supported by four separated foundations with a distance of 25 m (root), all of which were grouted with concrete, as shown in Figure 3 (b) and Figure 3 (c).
(a) The high of the towers
(b) Schematic diagram of tower foundation (c) One of the Foundations of the UHV Tower
Figure 3. Construction and characteristic of the Ultra High Voltage line tower
Point 3: It is not clear why the vertical line method is more accurate than the other one ? Please describe the reasons more clearly,
Response 3: I am very sorry not to explain the the reasons why the vertical line method is more accurate than the other one clearly. There are two reasons including the protective coal pillar shape and the amount of protective coal pillars.
(1) The range obtained by the vertical line method for each line tower was a quadrilateral. It meant the protective coal pillar area depended on every individual quadrilateral area. However, the range obtained by the linear structure method was a line parallel to the protected boundary delineated by the maximum size of the protective coal pillar. For example, the size of the protective coal pillar for N0. 1, N0. 2, N0. 3 were 100m, 120m, 130m respectively. Therefore, it was finally selected 130m to draw the line parallel to the protected boundary, which resulting in a waste of large amount of coal resources. In other words, the protective coal pillar shape should be many quadrilaterals instead of a line parallel to the towers.
(2) The total area and the total amount of protective coal pillar with the vertical line method were less than that with linear structure method, which created huge economic benefits.
In summary, the vertical line method is more accurate than the linear structure method.
Thanks!
Point 4: Author have not mentioned any limitations of the presented methods. Please consider this remark.
Response 4: I am very sorry not to mention the limitations of the presented methods. The limitations of the presented two methods are that the formula (3) and (4) in step 3 of section 4.1.1are easily calculated but a lot of repeated calculation. It should be careful to collect and analyze the data.
Thanks!
Point 5: The statement in the rows 260-262 are obvious. It is difficult to see the reasons given in the last sentence in this chapter (rows 262-264).
Response 5: Another statement was added in rows 260-262: In addition, according to Figure 9, the range obtained by the vertical line method for each line tower was a quadrilateral. It meant the protective coal pillar area depended on every individual quadrilateral area. However, the range obtained by the linear structure method was a line parallel to the protected boundary delineated by the maximum size of the protective coal pillar, which resulting in a waste of large amount of coal resources.
Thanks!

Reviewer 2 Report
Please improve some terms in technical English.
fig. 1 - please improve readability,
line 89 - south instead of north?
line 112 - table 3 instead of 2?
line 123 - "strike direction angle" and "head end angle" this is not English term,
line 150 - angle of critical deformation - this is not English term,
line 297 - please check writing, should be: Analytical model of protective...
Author Response
Dear Reviewer:
I appreciate it very much to receive your comments on my paper. I have revised the manuscript according to the reviewers' comments one by one carefully, as shown:
Point 1: fig. 1 - please improve readability,
Response 1: I rewrite what the fig. 1 indicated and improve the readability: Sihe Coal Mine is located in the North-west of Jincheng City, Shanxi Province, China. It is divided into two mining areas, including the West area and the East area, as shown in Figure 1. The study area of the paper is located in the East area of Sihe Coal Mine.
Thanks!
Point 2: line 89 - south instead of north?
Response 2: I check the location between seam #15 and line tower No. N157~N167 in fig. 2(b) . It shows that seam #15 is on the north instead of south of the line tower No. N157~N167. The line 89 is right.
Thanks!
Point 3: line 112 - table 3 instead of 2?
Response 3: I check what the line 112 said. It is actually table 3 instead of 2. I have revised it.
Thanks!
Point 4: line 123 - "strike direction angle" and "head end angle" this is not English term,
Response 4: I am very sorry not to explain these terms clearly and not to cite the detailed reference.
A note was added after Table 3 to explain these terms. Note: The strike direction angle means the angle along the seam strike direction; head end angle means the angle towards the head end direction; tail end angle means the angle towards the tail end direction.
It is cited by the reference “Guo W.B.; Bai E.H.; Ma X.C. Surface Subsidence Characteristics and Indices for Mining in “Three-Soft” Coal Seams. ICGCM China 2014 Proceedings (English volume), Beijing, China, 24 October; China University of Mining and Technology Press: Beijing, China, 2014, 161-165.” I have add these in the paper.
Thanks!
Point 5: line 150 - angle of critical deformation - this is not English term,
Response 5: I am sorry not to cite the detailed reference. This English term is cited by the book “ English Writing Method & Vocabulary in Coal Mine Ground Control”, as shown in Figure. 1. I have add the reference in the paper.
Thanks!
Figure. 1 The reference of the English term “angle of critical deformation”
Point 6: line 297 - please check writing, should be: Analytical model of protective...
Response 6: I check the line 297 writing carefully. It is actually “Analytical model of protective....” instead of “Aniayitcal model of proetetive...”. I have revised it.
Thanks!
